# Developing Consensus Standard Operating Procedures (SOPs) to Evaluate New Types of Insecticide-Treated Nets

**DOI:** 10.3390/insects13010007

**Published:** 2021-12-21

**Authors:** Natalie Lissenden, Jennifer S. Armistead, Katherine Gleave, Seth R. Irish, Jackline L. Martin, Louisa A. Messenger, Sarah J. Moore, Corine Ngufor, Natacha Protopopoff, Richard Oxborough, Angus Spiers, Rosemary S. Lees

**Affiliations:** 1Department of Vector Biology, Liverpool School of Tropical Medicine, Pembroke Place, Liverpool L3 5QA, UK; katherine.gleave@lstmed.ac.uk (K.G.); rosemary.lees@lstmed.ac.uk (R.S.L.); 2Innovation to Impact, Pembroke Place, Liverpool L3 5QA, UK; angus.spiers@innovation2impact.org; 3U.S. President’s Malaria Initiative, U.S. Agency for International Development, Washington, DC 20547, USA; jarmistead@usaid.gov; 4U.S. President’s Malaria Initiative, Entomology Branch, Division of Parasitic Diseases and Malaria, Center for Global Health, Centers for Disease Control and Prevention, Atlanta, GA 30329, USA; xjs7@cdc.gov (S.R.I.); louisa.messenger@lshtm.ac.uk (L.A.M.); 5Kilimanjaro Christian Medical University College, National Institute for Medical Research, Moshi P.O. Box 2240, Tanzania; lyimojaqueen@gmail.com; 6Vector Control Product Testing Unit, Environmental Health and Ecological Science Department, Ifakara Health Institute, Bagamoyo P.O. Box 74, Tanzania; smoore@ihi.or.tz; 7Nelson Mandela African Institution of Science and Technology, Arusha P.O. Box 477, Tanzania; 8Vector Biology Unit, Department of Epidemiology and Public Health, Swiss Tropical and Public Health Institute, Socinstrasse 57, 4051 Basel, Switzerland; 9Faculty of Science, University of Basel, St. Petersplatz 1, 4002 Basel, Switzerland; 10Department of Disease Control, London School of Hygiene and Tropical Medicine, London WC1E 7HT, UK; corine.ngufor@lshtm.ac.uk (C.N.); Natacha.Protopopoff@lshtm.ac.uk (N.P.); 11Centre de Recherche Entomologique de Cotonou, Cotonou BP 2604, Benin; 12PMI VectorLink Project, Abt Associates, Rockville, MD 20852, USA; Richard_Oxborough@abtassoc.com

**Keywords:** insecticide-treated net (ITN), PBO ITN, synergist ITN, dual-AI ITN, insecticide resistance management (IRM), method validation, durability monitoring

## Abstract

**Simple Summary:**

Malaria control relies on insecticide-based tools which target the mosquito vector. Predominantly, a group of insecticides called pyrethroids are used in these tools. Globally, however, mosquitoes are increasingly developing resistance to pyrethroids. Subsequently, new products, such as insecticide-treated nets (ITNs), which contain combinations of insecticides from different classes, or chemicals that work synergistically with pyrethroids, are being developed. Several of these new net types are being rolled out for testing and use. However, standardized methods to measure how long these nets remain active against mosquitoes are lacking, which makes evaluating the long-term efficacy of these products challenging. In this publication, we propose a pipeline used to collate and interrogate several different methods to produce a singular ‘consensus standard operating procedure (SOP)’, for monitoring the residual efficacy of three new net types: pyrethroid + piperonyl butoxide (PBO), pyrethroid + pyriproxyfen (PPF), and pyrethroid + chlorfenapyr (CFP).

**Abstract:**

In response to growing concerns over the sustained effectiveness of pyrethroid-only based control tools, new products are being developed and evaluated. Some examples of these are dual-active ingredient (AI) insecticide-treated nets (ITNs) which contain secondary insecticides, or synergist ITNs which contain insecticide synergist, both in combination with a pyrethroid. These net types are often termed ‘next-generation’ insecticide-treated nets. Several of these new types of ITNs are being evaluated in large-scale randomized control trials (RCTs) and pilot deployment schemes at a country level. However, no methods for measuring the biological durability of the AIs or synergists on these products are currently recommended. In this publication, we describe a pipeline used to collate and interrogate several different methods to produce a singular ‘consensus standard operating procedure (SOP)’, for monitoring the biological durability of three new types of ITNs: pyrethroid + piperonyl butoxide (PBO), pyrethroid + pyriproxyfen (PPF), and pyrethroid + chlorfenapyr (CFP). This process, convened under the auspices of the Innovation to Impact programme, sought to align methodologies used for conducting durability monitoring activities of next-generation ITNs.

## 1. Introduction

Globally, malaria control progress is plateauing, and, in some instances, case numbers are rising [1]. Although the reasons for this are multifaceted, an increasing and intense resistance to pyrethroids in *Anopheles* vectors is almost certainly a contributing factor. Insecticide-treated nets (ITNs) have significantly contributed to the control of malaria over the past two decades [2]. However, currently, all WHO-prequalified ITNs contain pyrethroids [3], and pyrethroid resistance is widespread in all major malaria vectors [4,5].

In response to growing concerns over the sustained effectiveness of solely pyrethroid-based control tools, new products are being developed and evaluated. Examples of these are dual-active ingredient (AI) ITNs containing an additional insecticide, or synergist ITNs which contain an insecticide synergist, in combination with a pyrethroid. These net types are often termed ‘next-generation’ insecticide-treated nets. The second AIs have a different mode of action (MoA) from their partner pyrethroid, to improve the control of resistant vector populations.

The current methods for measuring ITN durability [6] were developed for pyrethroid-only nets, which cause rapid knockdown and death in susceptible mosquitoes. Consequently, the different MoAs of the new insecticides necessitate the need for new protocols to reliably measure net durability. In nets with the synergist piperonyl butoxide (PBO), the PBO works by improving the efficacy of the pyrethroid it is paired with, in populations with pyrethroid resistance due to increases in oxidase activity, and is itself generally non-insecticidal. Without suitable mosquito strains or net controls, it is difficult to determine if the synergist component of the net is long-lasting using the currently recommended methods. For other AIs, such as chlorfeniapyr, which targets the insect mitochondria, or pyriproxyfen, which is a juvenile hormone analogue, ‘non-standard’ endpoints such as delayed mortality and insect fertility and fecundity need to be measured to assess biological durability (bioefficacy, measured through direct impact on mosquitoes).

Several of these new types of ITN are being evaluated in large-scale randomized control trials (RCTs) and pilot deployment schemes. These trials are expected to demonstrate the biological durability, attrition, and fabric integrity of these new net types when under long-term household use. Measuring the biological durability of the ITNs involves assessing the insecticidal activity of a sub-sample of randomly selected nets withdrawn from the field. There is an urgent need for methods to reliably measure the bioefficacy of these nets, to collect baseline data, and to subsequently measure the durability of biological efficacy of nets collected from the field after fixed periods of use. This has resulted in methods for measuring net bioefficacy and biological durability being developed and utilized by multiple programme teams, which makes comparing the results of these studies complex. A better approach would be for programme teams to adopt a single, standardized method validated using a multi-site approach.

In this publication, we demonstrate the process used to collate and interrogate several different methods to produce a singular ‘consensus standard operating procedure (SOP)’, for evaluating the biological efficacy of new net types, suitable for durability monitoring. Our objective was to create procedures that build on the experience from studies already underway. We also considered the feasibility of conducting these methods in as many sites as possible, accounting for factors such as throughput of mosquito colonies and space, which can preclude the use of certain methods and inform choices about sample sizes and replicate numbers.

This project forms part of a package of work to improve entomological methods in vector control and is supported by Innovation to Impact (I2I) at the Liverpool School of Tropical Medicine (LSTM). Three new types [7] of ITN are used as case studies: pyrethroid + piperonyl butoxide (PBO), pyrethroid + pyriproxyfen (PPF), and pyrethroid + chlorfenapyr (CFP). The final consensus SOPs for measuring the biological durability of these net types are included in Appendix A.

## 2. Materials and Methods

For each net type, a collaborative process of method development and iterative drafting was conducted to produce a consensus SOP (Figure 1). Initially, a group of stakeholders was formed. Inclusion in these groups was based on having (1) a research interest in the development or deployment of new net types, (2) experience in the development or testing of new net types, or (3) an involvement in ongoing trials or deployment schemes of new net types. Available methods for measuring the biological durability of each net type were then identified through consultations with stakeholder groups and literature searches. This was not a systematic process, and for each net type, several historical procedures exist which were not considered here. Rather, the focus was to identify SOPs currently being developed or utilized which evaluated the biological durability of new net types and to use them to align the methods on points of difference. For each net type, the experimental parameters of the method were established (i.e., exposure method, controls used, population, replicates, endpoints). Values for each parameter were extracted from all accessible methods and compared before a ‘consensus value’ was suggested for each experimental element. Other methodological questions were identified for discussion. At this stage, the method development document was shared with the stakeholder group for comment, and further discussed on a group call. The feedback on the method development was then used to prepare a draft consensus SOP. The draft was distributed with the group for a second round of comments and discussion. Following the incorporation of this feedback, a final consensus SOP was produced and submitted to the group for approval.

## 3. Case Study 1: ITNs Containing Pyrethroid plus Piperonyl Butoxide (Pyrethroid + PBO Nets)

Currently, six pyrethroid + PBO nets are prequalified by the WHO (DuraNet Plus, VEERALIN, PermaNet 3.0, Tsara Boost, Tsara Plus, Olyset Plus) [3]. These vary in several specifications (Appendix A) such as pyrethroid AI, PBO concentration, and location of PBO on the net (roof only or on all panels). A conventional cone test, followed by a tunnel test for those nets which fail to reach cone bioassay thresholds [8], is suitable for exposing mosquitoes to pyrethroid + PBO nets and monitoring mortality. Certain methodological parameters of the WHO cone test, such as replicate number and control nets, vary depending on if the assay is being used for WHOPES (the precursor to WHO prequalification) phase I, II, or III testing. The WHO guidance states “candidate LNs (nets) treated with insecticides with effects on mosquitoes that differ from those of pyrethroids may require proof of principle and new assays” [8]; however, guidance or thresholds on how to interpret PBO-synergism for biological durability monitoring is not available.

Nine methodologies that measure pyrethroid + PBO net biological durability were identified through searching the literature and contacting key stakeholders (Table 1). Of these, methods were accessible for six of them (published or provided on request). Of the remaining three, one study had not yet finalized its methods (ID = 7), one confirmed it was not conducting biological durability monitoring (ID = 8), and one did not have biological durability monitoring listed as an intervention endpoint on its clinical trial registry; the authors were contacted to confirm this, but they did not respond (ID = 9). Values for each methodological parameter were extracted from the accessible SOPs and a ‘consensus’ value suggested for each parameter (Table 2). It was established that one method (ID = 2) was an updated version of another (ID = 1), so study #2 was later excluded.

### 3.1. Other Methodological Considerations Identified


Date, temperature, relative humidity, test species/strain (including resistance profiles), and mosquito age (days) should always be recorded.Time of testing and light–dark cycle of test mosquitoes should be recorded.Nets and mosquitoes should be acclimatized to the temperature and humidity of the testing room for a minimum of 1 h before testing. This is critical if nets have been stored in a refrigerator or cold room.For mosquitoes collected as larvae from the field, details on the collection procedure, such as the number and distribution of collection sites, and mosquito-rearing conditions, should be recorded.Some pyrethroid + PBO nets have different pyrethroid concentrations on the sides and the roof and this should be considered in the data recording and interpretation. Therefore, it is important that net pieces are well labelled to establish if the sample is from the roof or sides, and data should be recorded per net piece. Though analysis should be pooled for each net for interpretation, having the data disaggregated in this way will allow for further interrogation of the data if required.


### 3.2. Changes Made to the Proposed Pyrethroid + PBO Methods following Stakeholder Discussions

It was decided that it was clearer to structure the SOP based on net panel type (i.e., a pyrethroid-only net panel), rather than describe testing based on nets with ‘PBO all over’ vs. ‘PBO mosaic net’ (PBO on the roof only). This structuring should allow adaptation to ITNs that may be developed in the future with different net panel configurations.Number of pieces sampled from each net: WHO biological durability monitoring [6] for pyrethroid-only nets recommended sampling one piece from the net roof and three–four pieces from the sides (four–five total). Our original proposal for pyrethroid + PBO nets was to sample three pieces from the roof and three from the sides (six total). The decision to test more roof samples was based on research which has shown greater mosquito activity on the net roof [9,10,11], the acknowledgement that some pyrethroid + PBO nets have different physio-chemical properties on the net roof, and that, during their manufacture, roof panels come from different net runs than side panels [12]. However, weighing up the benefits of a more precise measurement of intra-net heterogeneity by using six replicates per net against the challenge of evaluating large cohorts of ITNs with high numbers of mosquitoes per net, it was decided that the key measurement was the estimated bioefficacy of a cohort of ITNs. Therefore, it is important to be able to evaluate as many ITNs as possible (as nets have a high degree of heterogeneity due to different variability in use and care) while balancing this against the requirement for mosquitoes. Four samples per net (two from the roof, two from the sides) will allow the maximal numbers of samples to be tested without putting undue strain on testing facilities.Replicates: The original proposal was four replicates per net sample based on the WHOPES recommendations for pyrethroid-only nets [6]. However, this made the required mosquito numbers unfeasible. The consensus was that two replicates per net sample was sufficient. If mosquito numbers are abundant, testing should prioritize testing more nets (if available), as this will provide more precision. If additional nets are not available, surplus mosquitoes could be used to conduct more test replicates. After the consensus SOP was developed, a pre-print was published [13], which contained additional methods for the planned evaluation of the biological durability of PBO nets. The methods published in that report were compared to the draft consensus SOP and, methodologically, these were found to be largely the same, with some variability in sampling position and number of net samples/replicates.Testing should primarily use the WHO cone method specified in the consensus SOP (Appendix A). A tunnel test may be used as a second test when nets fail to meet WHO thresholds (<95% 60-min knockdown or <80% 24-h mortality in a susceptible strain [6]), although this is not preferred. Currently, there are no recommended thresholds for resistant mosquito strains. Following feedback from stakeholders, a final consensus SOP was produced and approved by the group (Appendix A.

## 4. Case Study 2: ITNs Containing Pyrethroid plus Pyriproxyfen (Pyrethroid + PPF Nets)

Royal Guard, developed by Disease Control Technologies, is currently the only WHO prequalification listed pyrethroid + PPF net (Appendix A). The WHO cone test is a suitable method for exposing mosquitoes to pyrethroid + pyriproxyfen (PPF) nets for measuring the nets’ biological durability, but different endpoints are needed for each active ingredient. Knockdown and mortality can be used to assess the bio-efficacy of the pyrethroid but the most suitable endpoints for PPF, a juvenile hormone analogue that affects fertility and fecundity in mosquitoes, need to be defined.

Seven documents detailing methods for evaluating pyrethroid + PPF nets were provided by stakeholders (Table 3). One of these (ID = 1) did not measure fertility endpoints. Of the remaining documents, four detailed methods for oviposition observations, and two detailed methods for ovary dissection.

To reach a consensus SOP for both methods, methodological parameter values were extracted from available SOPs and a ‘consensus’ value was proposed for each one (Oviposition: Table 4; Dissections: Table 5). Methods for both oviposition and dissection are included, as discussions showed differences in preference between labs for one or the other method (Figure 2).

### Changes Made to the Proposed Methods following Stakeholder Discussions

The option to score oviposition and then dissect those that did not lay was discounted. This would have meant dissections were being conducted on non-standardized days, making results incomparable to data collected using the standard dissection method, and likely resulting in a small sample size for that subset. For similar reasons, those which died before oviposition counts should not be dissected and scored.As we do not expect the pyrethroid to impact fertility, and we are using a pyrethroid-resistant strain, the untreated net is a useful negative control, and oviposition inhibition can be compared to this. Therefore, the decision was made not to include a pyrethroid-only net.Questions remain regarding the ‘net effectiveness threshold’ for sterility endpoints. For pyrethroid only nets, a net is considered effective if KD_60_ is >95% or 24-h mortality is >80% [6]. We do not yet know what an operationally meaningful level of sterility is, i.e., what level of sterility in a cone test means the net is controlling mosquitoes in the field. Hence, it is not yet possible to set a threshold for biological durability monitoring, and the best approach is to simply monitor for a reduction in sterilizing effect over time. However, this question is critical and should be considered as data is generated.When analyzing the results, the untreated net and the test net should be paired, i.e., a single control for the day acts as the benchmark for all tests on that day, and inhibition is calculated against that day’s control. Inhibition can be calculated by odds ratio using regressions.Following the development of the consensus SOP, a pre-print was published, which contained additional methods planned for evaluating biological durability of PPF nets [13]). These methods were compared to the drafted consensus SOP and found to be methodologically the same, apart from some variability in sampling position and number of net samples/replicates. Following feedback from stakeholders, a final consensus SOP was prepared and approved by the group (Appendix A.

## 5. Case Study 3: ITNs Containing Pyrethroid plus Chlorfenapyr (Pyrethroid + CFP Nets)

Interceptor G2 (IG2), developed by BASF, is currently the only WHO prequalification listed pyrethroid + CFP net (Appendix A). The cone test has been shown to be ineffective in reliably measuring the bioefficacy of the chlorfenapyr component of IG2 nets [16], and so an alternative bioassay is needed. There is a growing consensus around the WHO tunnel test as being the best method to assess IG2 bioefficacy. This should be run in parallel with a standard WHO cone test [6], which assesses the biological durability of the alpha-cypermethrin component of the net. The SOP discussed and included (Appendix A) here is related to assessing the biological durability of the CFP component.

Eight documents, detailing methods used for evaluating pyrethroid + CFP nets, were provided by stakeholders (Table 6). Of these, three were generic SOPs for conducting the ‘net in tube’ cylinder assay (ID = 6) or tunnel test (ID = 7, 8), and did not contain specific experimental parameters for testing CFP nets, and, therefore, information was not extracted from them for comparison. Methodological parameters were extracted from the available SOPs, compared, and used to propose a ‘consensus’ value for each (Table 7).

### Changes Made to the Proposed Pyrethroid + CFP Methods following Stakeholder Discussions

Where tunnel testing is not possible, it would be beneficial to have an additional method available. It was established that S. Moore will be validating the I-ACT method [18] for IG2 testing, and K. Gleave will be validating the ‘Net in Tube’ (cylinder) test. When complete, we will include these SOPs with the tunnel-test methodology on the I2I website (https://innovationtoimpact.org/workstreams/methods-validation/). Accessed on 20 December 2021.Following a preliminary discussion with all stakeholders, a sub-group was formed with key individuals to start a draft proposal for the CFP methodology. In the initial meeting, representatives of BASF joined to share information on Interceptor G2. Following on from these discussions, a draft method development with methodological parameters for the tunnel test was shared with the sub-group, and this was refined before sharing with the full stakeholder group for approval.From a biological durability perspective, it was decided that it was not necessary to have a comparison to a new Interceptor net (IG1) and a new Interceptor G2 net (IG2) at every time point. Thus, these were removed as daily controls. Instead, the resistant strain should be characterized against the Ais in parallel with each round of bioassays, as recommended in Lees et al. (in prep), to investigate the additional effect of chlorfenapyr, and to confirm pyrethroid resistance and chlorfenapyr susceptibility to check that they have not drifted in the test strain during the test period.There is a lack of data on how mortality in tunnel tests changes with mosquito numbers (the standard is 100 mosquitoes in a tunnel). Reducing the sample to 50 mosquitoes per tunnel allows us to increase the sample pieces tested per net without increasing mosquito numbers. However, this also increases the risk of having to disregard testing results if high control mortality is observed—control mortality would still be based on 100 mosquitoes, but over two net replicates.Data comparing the use of 50 vs. 100 mosquitoes in tunnels with pyrethroid nets are available (Moore, Personal communication), and these data were considered to confirm the number of mosquitoes tested.Further to this, preliminary work to compare 50 vs. 100 mosquitoes in tunnels using Interceptor net and Interceptor G2 nets was conducted, and found no significant difference in these two numbers (Kamande, Personal communication).The number of mosquitoes required must be balanced against the number of replicates, since maximizing the number of nets, to measure efficacy of the ITN population, is key. There was some disagreement over which was the best balance. It is likely that the capacity to test more mosquitoes per net will be related to mosquito availability in the testing sites. Therefore, it is suggested we validate with the lower number to make the SOP less onerous for testing sites. We are interested in measuring the biological durability of the ITN population—not individual nets, which could be highly variable. Currently, the WHO recommends 30 nets per time point, but increasing this will provide better data. Thirty nets should be seen as the minimum. Reducing the number of mosquitoes may allow increases in replication to be possible.Control thresholds: blood-feeding must be >50% on the untreated control net. Mortality will be measured up to 72 h, due to the slow-acting nature of chlorfenapyr. Mortality in the untreated control must be <10% after 24 h and <20% at 72 h (both must be true for the test to be valid). Following feedback from stakeholders, a final consensus SOP was produced and approved by the group (Appendix A.

## 6. Discussion

Methodological consistency is crucially important when monitoring the durability of new net types, due to there not being validated methods to assess these tools. Even small differences in testing methods may lead to additional sources of variation in endpoints, making results difficult to interpret between countries, studies, and test facilities. The use of standardized testing methods streamlines the process of product evaluation, leads to a more rapid generation of consistent performance data across studies, and subsequently speeds up product uptake. In vector control, methods for new tools with novel modes of action are often developed in one site or by one group in response to a specific product or research question. This can narrow the applicability of that method, make it challenging to adopt it at other sites, or it may not be applicable to all products within a particular product class.

Developing evaluation methods in a collaborative group (‘consensus’ SOPs) allows the process to benefit from the collective knowledge and experience of a diverse set of stakeholders, and maximizes the chances for a specific methodology that will be widely relevant. However, developing a consensus SOP is just one of the first steps in the method-validation pipeline. Defining and improving the robustness of a method can be viewed as an incremental process which follows a stepwise progression from singular SOPs to consensus SOPs, to consensus SOPs that are experimentally validated at one site, and finally, to consensus SOPs that are validated at multiple sites. In this publication, we have defined the desired endpoints, and designed and refined methodologies for evaluating the biological durability of three new net types. The next steps in this process will be to (1) quantify inherent errors in the methods, (2) evaluate the ability of the methods to accurately characterize the vector control product, and (3) validate these results in multiple facilities. The scope of this would include assessing the methods’ ability to measure the biological durability of different products within the class of nets, and against different vector species. More information is gathered when a method is in operational use, which can help to improve or refine the method. At this stage, it is imperative to ascertain that the methods can be implemented and used successfully within research teams, and identify training needs, if required. This is to ensure that data collected using these methods are as transferable and comparable as possible.

The agreement on key entomological endpoints to be measured, followed by the use of standardized and validated methods to measure them, needs to be partnered with an acceptance of the need for flexibility in product evaluation. For instance, the SOPs developed here have been formulated based on nets that are currently in development/on the market and therefore may be unsuitable for new formulations or designs within the same product classes. However, it should be noted that this is the way that previous ITN guidelines were developed—in response to new technologies coming to market [6]. It is challenging to ‘future-proof’ methods from the outset, especially in a rapidly evolving landscape which must be sensitive to the pressures of evolving and emerging insecticide resistance. Therefore, the process cannot be averse to change or updates in the future, which would lead to stagnation in innovation and delayed decision making—such has been the situation with non-pyrethroid products being evaluated with tests designed for pyrethroids. Regular updates of guidance based on consensus among key stakeholders will harmonize data collection procedures and, ultimately, hasten progress towards the goal of bringing new vector control products to market more rapidly, using robust data-driven decision making.

To take this further, the dissemination of up-to-date methods is crucial to ensure relevant data are being collected whenever possible. This process, convened under the auspices of the Innovation to Impact programme, sought to align methodologies used by those conducting durability monitoring activities of new net types (so-called ‘next-generation ITNs’). While this objective was largely achieved through the engagement and insight of those involved, it is important to recognize that even though this process involved the key stakeholders in designing and implementing durability monitoring, the current durability monitoring guidelines [6] for these products may differ or simply do not exist. There is a clear need for further engagement with normative (WHO and control programmes) and implementation groups (Roll Back Malaria and others) to ensure up-to-date guidance for durability monitoring is available to all who may wish to access it.

## Figures and Tables

**Figure 1 insects-13-00007-f001:**
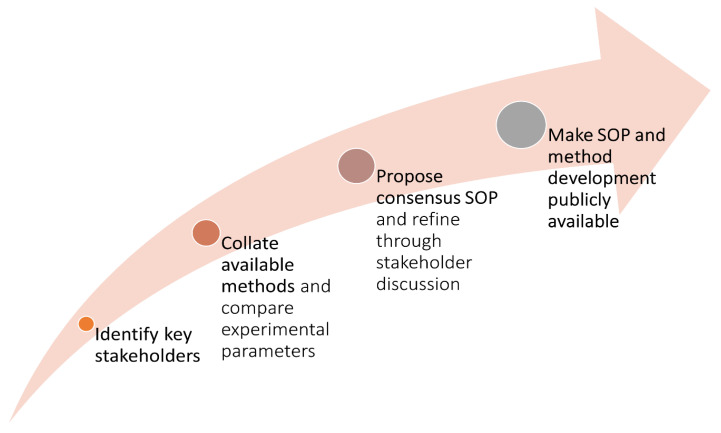
Infographic showing the process of method development used for producing consensus SOPs for biological durability monitoring of new net types.

**Figure 2 insects-13-00007-f002:**
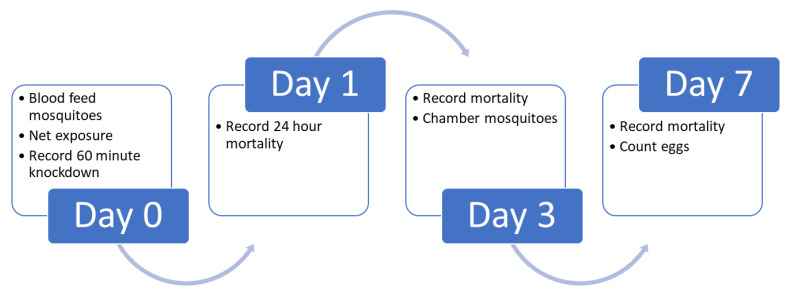
Infographic showing the methodological process for measuring sterility via scoring oviposition using chambering, following exposure to pyrethroid + PPF nets.

**Table 1 insects-13-00007-t001:** List of identified methods/trials measuring pyrethroid + PBO net biological durability.

ID	Contact	Biological Durability Monitoring	Method Availability
#1 PMI VectorLink SOP for NNP	Stephen Poyer, PSI	Yes	Provided
#2 NNP Burkina Faso DM protocol	Stephen Poyer, PSI	Yes	Provided
#3 LLINEUP trial Uganda	Amy Lynd, LSTM	Yes	Provided
#4 LLINEUP trial LSTM	Frank Mechan, LSTM	Yes	Provided
#5 Nigeria trial (Awolola et al., 2014)	Samson Awolola, NIMR	Yes	Published
#6 Kenya SMART Trial NCT04182126	Guiyun Yan, UC Irvine	Yes	Provided
#7 ISRCTN99611164	David Weetman, LSTM	Yes	Method not set
#8 JPRN-UMIN000019971	Noboru Minakawa, Nagasaki University	No	-
#9 NCT03289663	Gillon Ilombe, University of Kinshasa	Unclear	-

Abbreviations: DM = Biological durability monitoring; LSTM = Liverpool School of Tropical Medicine; NIMR = Nigerian Institute of Medical Research; NNP = New Nets Project; PMI = President’s Malaria Initiative; PSI = Population Services International; SOP = Standard operating procedure.

**Table 2 insects-13-00007-t002:** Methodological parameters extracted from pyrethroid + PBO net biological durability monitoring methods. Methods were compared and a consensus value was proposed for each parameter for discussion by the stakeholder group. Justification for this choice regarding each parameter is listed. Superscript numbers = Study ID.

	PMI VectorLink SOP ^1^	LLINEUP SOPs ^3^	Mechan PhD Project ^4^	Nigeria Trial ^5^	Kenya SMART Trial ^6^	Proposed for Consensus SOP	Justification
Author	PMI VectorLink	Lynd (LSTM)	Mechan (LSTM)	Awolala (Nigeria medical institute)	Yan (University of California)	Lees and Lissenden (LSTM)	-
Method of exposure (primary test)	Cone (3 min)	Cone (3 min)	Cone (3 min)	Cone (3 min)	Cone (3–5 min)	Cone (3 min)	This is the standard exposure time used in WHO cone bioassays [6].
Controls	Untreated net. New pyrethroid-only net. New pyrethroid + PBO net.	Untreated net control.	-	-	Untreated net control.	Negative control: Untreated control net. Positive control 1: New pyrethroid + PBO net of the same brand. Positive control 2: Pyrethroid-only net of the same pyrethroid (as similar as possible).	Untreated net controls for handling procedure and checks for contamination. New pyrethroid + PBO net provides ‘baseline’ mortality and allows us to monitor the suitability of test mosquito strains. New pyrethroid-only net controls for the mortality conferred by the pyrethroid product.
Age of mosquito	-	3–5 days	3–5 days	2–3 days	2–5 days	2–5 days	Age range recommended for bioefficacy testing [6]. It encompasses the age ranges previously tested and is logistically feasible.
Mosquitoes per rep	5	5	5	5	10	5	This is the standard number used in WHO cone bioassays [6]
Samples per net	PBO all over: 4 pieces (1 roof); PBO roof only: 6 pieces (3 roof).	2 pieces from the top of the net (though 3 pieces were cut from net).	2 pieces from the top of the net (25 × 25 cm^2^).	5 pieces (1 top, 4 sides).	5 pieces (1 top, 4 sides); 30 × 30 cm^2^.	4 pieces (2 from net roof, 2 from net sides).	This aligns with the other new net type SOPs, and with the standard WHO biological durability testing where (post-baseline) 4 pieces of net are tested [6]. The decision to take equal pieces from the roof is due to greater mosquito activity observed here [9,10,11] and because some nets only have PBO on the roof. During their manufacture, roof panels can come from different net runs than side panels [12].
Replicate tests per piece of net	2 cones per net piece; PBO all over: *n* = 40; PBO roof only: *n* = 60.	25 per piece (*n* = 50).	3 cones simultaneously on each piece of net (6 cones total, *n* = 30).	1 cone per piece (25 mosquitoes).	2 cones per rep (*n* = 100 mosquitoes).	2 replicates per piece (8 cones per net).	Likely to be a feasible number for testing. Numbers will be finalized during multicenter validation of the SOP.
Replicate nets per treatment	-	-	-	+30 (35 houses selected).	18 nets	A minimum of 30 nets of each treatment at each time point.	WHO guidelines [6] recommend a minimum of 30 nets (at time points 0–24 months), and a minimum of 50 nets at 36 months testing.
Species/strain	A pyrethroid-susceptible and a pyrethroid-resistant strain.	-	A pyrethroid-susceptible (*An. gambiae* Kisumu) and a pyrethroid-resistant strain (*An. gambiae* Busia).	A pyrethroid-susceptible strain (*An. gambiae* Kisumu).	A pyrethroid-susceptible strain (*An. gambiae* Kisumu).	Lab-reared pyrethroid-susceptible strain. Lab-reared pyrethroid-resistant strain. Lab strains characterized before and after the bioassays for each time point, as per strain characterization guidelines (Lees et al. in prep).	The susceptible strain is used to monitor the biological durability of the pyrethroid over time. The pyrethroid-resistant strain is used to monitor the impact of PBO over time.
Storage of netting pieces (prior to testing)	-	Room temperature	Refrigerator-stored (5 °C)	-	In foil (4 °C)	Refrigerated or in a cool dry place, at <5 °C or as per manufacturer’s instructions.	-
Entomological endpoints measured	Knock down (KD): 60 min; Mortality: 24 h.	KD: 60; Mortality: 24 h + alive with 2 or less legs, and the number alive and flying well with 3 or more legs.	KD: 60 min. Mortality: 24 h.	-	KD: 10, 20, 30, 40, 50, and 60 min. Mortality: 24 h.	KD: 1 h. Mortality: 24 h.	These endpoints are sufficient to capture the efficacy of a pyrethroid + PBO net.

**Table 3 insects-13-00007-t003:** List of identified methods/trials measuring pyrethroid + PPF net biological durability.

ID	Contact	Biological Durability Monitoring	Method Availability
#1 CNRFP tunnel test AvecNet	Emile Tchicaya, CSRS	Yes	N/A
#2 LSTM Cone test AvecNet (Toé et al., 2019)	Hyacinth Toé, CNRFP	Yes	Provided
#3 Oviposition SOP, CREC, Benin	Corine Ngufor, LSHTM	Yes	Provided
#4 Dissection SOP, CREC, Benin	Thomas Syme, LSHTM	Yes	Provided
#5 Dissection SOP, KCMUCO, Tanzania	Jackline Martin, KCMUCo	Yes	Provided
#6 Royal Guard Trial [14]	Corine Ngufor, LSHTM	Yes	Provided
#7 WHO PPF DC bottle study	Vincent Corbel, IRD	Yes	Provided

Abbreviations: CNRFP = Centre National de Recherche et de Formation sur le Paludisme; CREC = Centre de Recherche Entomologique de Cotonou; DC = Diagnostic concentration; IRD = Institute of Research for Development; KCMUCo = Kilimanjaro Christian Medical University College; LSHTM = London School of Hygiene and Tropical Medicine; LSTM = Liverpool School of Tropical Medicine; PPF = Pyriproxyfen; SOP = Standard operating procedure.

**Table 4 insects-13-00007-t004:** Methodological parameters extracted from pyrethroid + PPF net biological durability monitoring methods, which scored mosquito oviposition. Methods were compared and a consensus value was proposed for each parameter. Justification for this choice regarding each parameter is listed. Superscript numbers = Study ID.

	Toé et al., 2019, Malaria Journal ^1,2^		CREC, Benin SOP/BL/131/03-S ^3^	Ngufor et al., 2020 Scientific Reports ^6^		WHO SOP ^7^	Proposed for Consensus SOP	Justification
Author(s)	Toé, Tchicaya, Ranson, Morgan, and Grisales		Gregbo, Fagbohou, and Ngufor	Ngufor		Corbel (based on LITE SOP)	Lees and Lissenden	-
Method of exposure	Cone Test	Tunnel Test (nets that did not reach target in cone test).	SOP-only covers post-exposure.	Cone Test	Tunnel Test	TGAI on bottles	Cone Test	The cone test has been used in several studies to evaluate PPF nets and seems to be a suitable method of exposure.
Exposure time	3 min	15 h, 18:00–09:00 h.	-	3 min	Overnight	1 h	3 min	This is the standard exposure time used in WHO cone bioassays [6]. Preliminary validation testing will be conducted to look at effect of exposure time.
Controls	Untreated net (4 reps per day, *n* = 20 mosquitoes). PPF-only net (4 reps per day, *n* = 20 mosquitoes).	Untreated netting.	-	Royal Sentry (alpha-cypermethrin net). Untreated control net.	Royal Sentry (alpha-cypermethrin net). Untreated control net.	Does not state treatment of control bottles.	Negative control: Untreated control net. Positive control: New pyrethroid + PPF net of the same brand.	Untreated net controls for handling procedure and checks for contamination, and provides denominator for measuring oviposition inhibition. New pyrethroid + PPF net provides ‘baseline’ and allows us to monitor the suitability of test mosquito strains.
Species/strain	Kisumu (pyrethroid-susceptible) in CNRFP, Kisumu and Tiassalé 13 (pyrethroid-resistant) in LSTM. Sterilizing effect only tested in LSTM on Tiassalé 13 that survived the Cone Test.	Kisumu	-	Kisumu and pyrethroid-resistant *An. gambiae* Cove strain.	Kisumu and pyrethroid-resistant *An. gambiae* Cove strain.	Susceptible strains of each species.	Lab-reared pyrethroid-susceptible strain Lab-reared pyrethroid-resistant strain Lab strains characterized before and after the bioassays for each time point as per strain characterization guidelines (Lees et al. In prep).	Lab-reared strains increase the likelihood of forced oviposition, yielding high rates. Pyrethroid-susceptible strain to monitor pyrethroid durability. Pyrethroid-resistant strain to monitor durability of PPF.
Age of mosquitoes	3–5 days	5–8 days	-	2–5 days old	5–8 days	5–7 days old, fed and inseminated.	3–5 days	This age range falls within the range of standard cone test (2–5 days, [6]) but allows an extra day for mating to increase likelihood of insemination. Effect of age for PPF is unknown and could be validated, but should be held constant until it is.
Mosquitoes per replicate	5	100	-	5	~80	25/bottle, 2 bottles/concentration, equal numbers of controls.	5 per cone.	This is the standard number used in WHO cone bioassays [6].
Samples per net	3 panels per net, one from each side at CNRFP, and 4 further panels for LSTM. 4 tests per panel at CNFRP, 3 further panels in LSTM.	‘Nets that did not reach the target’.	-	1	1	-	4 pieces from each net. Two from the roof, two from the sides.	This aligns with the other next-gen net SOPs, and with the standard WHO durability testing where (post-baseline) 4 pieces of net are tested [6]. The decision to take equal pieces from the roof is due to greater mosquito activity observed here [9]. During their manufacture, roof panels can come from different net runs than side panels [12]
Replicate tests per piece of net	3	?	-	1	1	N/A	2 replicates per piece (8 cones per net).	Consensus was that this was a feasible number for testing. Numbers will be confirmed during multi-center validation.
Replicate nets per treatment	24 of each type, or as many as available (high attrition), per timepoint.	‘Nets that did not reach the target’.	-	4 (2 control)	3	N/A	A minimum of 30 nets for each treatment at each time point.	WHO guidelines [6] recommend a minimum of 30 nets (at time points 0–24 months), and a minimum of 50 nets at 36 months testing.
Blood feeding timing	24 h post-exposure (LSTM: 30 min blood meal using Hemotek membrane feeding system).	-	Before exposure.	Before exposure (separate group b/d after exposure failed to feed and too few survived).	Unfed females used in test. Only blood-fed during tunnel were measured after for sterilizing effects.	Fed in the hour before exposure.	3–9 h before net exposure Blood fed using method of feeding standard for the test population (e.g., Hemotek membrane feeding system, arm feed, animal fed to repletion).	There is little data available and some contradiction on the impact of time of blood feeding, and this could be validated. Consensus was that this was a suitable and logistically possible method.
Timing of chambering	24 h post-exposure (LSTM) 72-h post-bloodmeal, 96-h post-exposure	Sterilizing effect not measured.	-	-	-	72 h post-exposure (73 h post b/m).	72 h post-exposure (Day 3).	This allows 3 days for bloodmeal development and egg maturation.
Method of chambering	30-mL cell culture tubes, moist cotton wool, and filter paper, individuals. Chambered for 3 days.	-	Cup, 50 mL water, 10% glucose cotton wool, individuals.	Individuals	-	100-mL plastic cups, 30 mL water, 10% glucose, individuals.	The chambering equipment used (i.e., culture tubes or plastic cups) is not critical and should reflect what method each lab has capacity to conduct. The same setup should then be used for all treatments and replicates. When oviposition in the untreated control is <20%, test results should be discarded and repeated.	20% oviposition threshold in the untreated control is based on power calculations performed by Joe Wagman (PATH).
Entomological endpoints measured	KD: 60 min; Mortality: 24 h; Number blood-fed; Eggs laid per female; Number 2nd instar larvae per female; Oviposition rate, fecundity, hatch rate, and fertility.	Blood-fed and dead after test.	Daily mortality to day 8. Count eggs and larvae on day 4 and day 8.	KD: 60 min. Mortality: 24-h mortality, individual oviposition: % reduction in oviposition rate, % reduction in fecundity, % reduction in offspring.	# alive/dead and # fed/unfed in each section, 24-h mortality, individual oviposition: % reduction in oviposition rate, % reduction in fecundity, % reduction in offspring.	KD: 60 min. Daily mortality (pre- and post-chambering until Day 8). Presence of eggs on day 8 post-exposure. Oviposition rate. Oviposition inhibition.	Primary endpoint: oviposition inhibition (calculated compared to untreated control. Additional measures: KD: 60 min, 24-h mortality, 72-h mortality (when chambering). Oviposition (egg laying) counted on Day 7 post-exposure only (4 days post-chambering).	A preliminary validation test will be conducted to establish if other endpoints should be included, e.g., median number of eggs laid.
Length of bioassay	-	15 h	8 days post-exposure.	-	-	8 days post-exposure.	8 days (Day 0 = day of exposure).	-
Notes on the protocol	High-performance liquid chromatography (HPLC) conducted on net samples—3 samples from each of 4 panels. Sterilizing effect measured in rounds 1–5 (1–24 m).	Untreated control run for each round.	No food provided to eggs/larvae. Water with eggs transferred to larvae cup on day 4.	-	-	Test rejected if control mortality is 20% or more, or oviposition in controls is <30%.		-
Storage of netting pieces (prior to testing)	-	-	-	-	-	-	Refrigerated or in a cool dry place, but at <5 °C or as per manufacturer’s instructions.	-

**Table 5 insects-13-00007-t005:** Methodological parameters extracted from pyrethroid + PPF net biological durability monitoring methods, which scored ovary development following dissection. Methods were compared and a consensus value was proposed for each parameter. Justification for this choice regarding each parameter is listed. Superscript numbers = Study ID.

	CREC, Benin SOP BL/159/01-S v01 ^4^	KCMUCO, Tanzania SOP 008v02 ^5^	Proposed for Consensus SOP	Justification
Author	Syme	Martin, Matowo, and Furnival-Adams	Lees and Lissenden	-
Method of exposure	Not included in SOP	Cone test	Cone test	The cone test has been used in several studies to evaluate PPF nets and seems to be a suitable method of exposure.
Exposure time	Not included in SOP	3 min	3 min	This is the standard exposure time used in WHO cone bioassays [6]. Preliminary validation testing will be conducted to look at effect of exposure time.
Age of mosquitoes	Unknown	2–5 days old	3–5 days	This age range falls within the range of standard cone test (2–5 days, [6]) but allows an extra day for mating to increase likelihood of insemination. Effect of age for PPF is unknown and could be validated, but should be held constant until it is.
Blood feeding timing	‘Blood-fed at the time of collection/testing’.	Females ‘freshly blood fed’ for exposure.	3–9 h before net exposure. Blood fed using method of feeding standard for the test population (e.g., Hemotek membrane feeding system, arm feed, animal feed).	There is little data available and some contradiction on the impact of time of blood feeding, and this could be validated. Consensus was that this was a suitable, and logistically possible, method.
Mosquitoes per replicate	N/A	5	5 per cone	This is the standard number used in WHO cone bioassays [6].
Replicates per piece of net	N/A	20–25 replicates (*n* = 100–150); 4 per piece; 30 nets per treatment.	2 replicates per piece (8 cones per net)	Consensus was that this was a feasible number for testing. Numbers will be confirmed during multi-center validation.
Replicate nets per treatment	N/A		A minimum of 30 nets of each treatment at each time point.	WHO guidelines [6] recommend a minimum of 30 nets (at time points 0–24 months), and a minimum of 50 nets at 36 months testing.
Species/strain	*Anopheles* mosquitoes (generic SOP for dissection).	*An. gambiae* s.s. Muleba kis (kdr east and mixed-function oxidize resistance), or wild blood-fed resistance mosquitoes of unknown age with species id at time of dissection.	Lab-reared pyrethroid-susceptible strain. Lab-reared pyrethroid-resistant strain. Lab strains characterized before and after the bioassays for each time point as per strain characterization guidelines (Lees et al. in prep).	Pyrethroid-susceptible strain to monitor pyrethroid durability. Pyrethroid-resistant strain to monitor durability of PPF.
Time of dissection	72 h post-exposure	72 h post-exposure	72 h post-exposure	This allows 3 days for bloodmeal digestion and egg maturation.
Blinded samples	No	Yes	Yes	Controls for scorer subjectivity.
Number of scorers	2, in case of discrepancy calculate the average (only for egg count).	2, using slide or photograph if slide cannot be counted on the same day. 3 scorers in case of discrepancy.	2, using slide or photograph if slide cannot be counted on the same day. 3rd scorer in cases of discrepancy.	Controls for scorer subjectivity.
Microscope details	Can use dissecting microscope, better a compound microscope at 4× or 10×.	0.7× magnification, stereomicroscope.	Microscope details not critical. However, we recommend using a magnification of ×4 or ×10 for dissections and ×40 for observation of eggs.	-
Entomological endpoints measured	Live/dead and gravid/semi-gravid at time of collection, egg development stage, and fertility status of each mosquito, total number of eggs present in ovary (1/2 per female?).	KD: 60 min. Mortality: 24 h. Mortality: 48 h. Mortality: 72 h. % of dissected females with under-developed ovaries 72 h post-feeding. Proportion of dissected females with deformed eggs. Average number of eggs in the ovaries 72 h post-feeding.	Primary endpoint: Fertility inhibition (fertility rate/fertility rate in the negative control). Additional measures: KD: 60 min. 24-h mortality. Egg development stage. Fertility rate (proportion with developed ovaries/total).	A preliminary validation test will be conducted to establish if other endpoints should be included, e.g., number of eggs in each dissected ovary.
Definition of Fertility	Christophers’ scale to score development stage of eggs (I–V); female is fertile if eggs are V and sterile if eggs are I–IV.	Christophers’ stages to score development stage of eggs (I–V); female is fertile if eggs are V and sterile if eggs are I–IV. Inconclusive if both are present.	Score development stage of eggs (1–5) [15]. Female is classed as fertile if all eggs are 5 and sterile if eggs are 1–4. If both classes 4 and 5 are present, the results are inconclusive.	This is a well-established method for scoring fertility
Controls	-	Untreated net. Standard LN: Interceptor.	Negative control: Untreated control net. Positive control 1: New pyrethroid + PPF net of the same brand.	Untreated net controls for handling procedure and checks for contamination, and provides denominator for measuring oviposition inhibition. New pyrethroid + PPF net provides ‘baseline’ and allows us to monitor the suitability of test mosquito strains.
Notes on the protocol	Dissect all mosquitoes left alive at 72 h post-collection, but if there are not adequate numbers, also dissect dead mosquitoes at this time. Photographs taken of eggs.	Method from Detinova et al. 1962. Photographs taken of eggs.	If the testing site has the capacity to photograph dissected ovaries, then this should be conducted. Photographs can then be used in future training, and machine learning activities.	

**Table 6 insects-13-00007-t006:** List of identified methods/trials measuring pyrethroid + CFP net biological durability.

ID	Contact	Biological Durability Monitoring	Method Availability
NNP Burkina Faso DM (ID = 1)	Richard Oxborough, PMI	Yes	Provided
Tanzania cRCT (Martin et al., 2021) (ID = 2)	Jackline Martin, KCMUCo	Yes	Published pre-print
Net in tube CFP, LSTM (ID = 3)	Katherine Gleave, LSTM	Yes	Provided
PMI CFP Tunnel SOP (ID = 4)	Richard Oxborough, PMI	Yes	Provided
Residual efficacy of Interceptor G2 (ID = 5)	Seth Irish, CDC, and Richard Oxborough, PMI	Yes	Provided
PAMVERC SOP for cylinder assay (ID = 6)	Leslie Choi, LSTM	Yes	N/A, generic SOP
IT LN SOP 002 V04—Tunnel Tests (ID = 7)	Sarah Moore, IHI	Yes	N/A, generic SOP
CREC SOP.BL.112.05.S—Tunnel tests (ID = 8)	Corine Ngufor, LSHTM	Yes	N/A, generic SOP

Abbreviations: CFP = Chlorfenapyr; cRCT = Cluster Randomized Control Trial; CREC = Centre de Recherche Entomologique de Cotonou; IHI = Ifakara Health Institute; KCMUCo = Kilimanjaro Christian Medical University College; LSHTM = London School of Hygiene and Tropical Medicine; LSTM = Liverpool School of Tropical Medicine; NNP = New Nets Project; PAMCERC = Pan-African Malaria Vector Research Consortium; PMI = Presidents Malaria Initiative; SOP = Standard operating procedure.

**Table 7 insects-13-00007-t007:** Methodological parameters extracted from pyrethroid + chlorfenapyr net biological durability-monitoring methods. Methods were compared and a consensus value was proposed for each parameter. Justifications for this choice, regarding each parameter, are listed. Abbreviations: IG1 = Interceptor Net, Alpha-cypermethrin net; IG2 = Interceptor G2, Chlorfenapyr + Alpha-cypermethrin net. Superscript numbers = Study ID.

	NNP Burkina Faso DM ^1^	Tanzania cRCT ^2^			Net in Tube, LSTM ^3^	PMI SOP ^4^	Irish and Oxborough SOP ^5^	Proposed for Consensus SOP	Justification
Author(s)	NNP	JL Martin et al.			Irish, Oxborough & Gleave	PMI	Irish and Oxborough	Lissenden	
Method of exposure (primary test)	Cone test	Cone test	Tunnel test	Cylinder test	Cylinder test	Tunnel Test	Tunnel Test	Tunnel Test	The tunnel test has been used in several studies to evaluate CFP nets and seems to be a suitable method of exposure.
Exposure time	-	3 min	12–15 h	3, 15, 30, 60 min, ‘as necessary’	30 min		12–15 h	12–15 h	This is the standard exposure time used in WHO tunnel tests [6].
Controls	No exposure control	Untreated net IG1 collected at same time point.	Untreated net IG1 collected at same time point.	-	Untreated net Alphacypermethrin net (100 mg/m^2^).	Negative control New IG1 New IG2	Untreated net. New IG1. New IG2 (used up to 10 times).	Untreated net (Used up to 10 times). Untreated Control thresholds: blood-feeding must be >50%. Mortality must be <10% after 24 h and < 20% at 72 h. New IG1 and IG2 should be used to characterize strain prior to testing.	Untreated net controls for handling procedure and checks for contamination and provides denominator for measuring oviposition inhibition. New IG1 + IG2 nets provides ‘baseline’ and allows us to monitor the suitability of test mosquito strains.
Species/ strain	Pyrethroid-susceptible strain. Pyrethroid-resistant strain.	A pyrethroid-susceptible strain (Kisumu).	A pyrethroid-susceptible strain (Kisumu—failed cone nets only). Pyrethroid-resistant strain (Muleba-kis), regularly selected and profiled.	-	Pyrethroid-resistant strain (<70% mortality).	Pyrethroid-susceptible (Kisumu) strain Pyrethroid-resistant (VKPER) strain	Profiled pyrethroid-resistant strain (<70% mortality to new IG1).	Lab-reared pyrethroid-susceptible strain. Lab-reared pyrethroid-resistant strain. Lab strains characterized before and after the bioassays for each time point, as per strain characterization guidelines (Lees et al. In prep).	The susceptible strain is used to monitor the biological durability of the pyrethroid over time. The pyrethroid-resistant strain is used to monitor the impact of CFP over time.
Age of mosquito	2–5 days	2–5 days	-	-	3–5 days	-	5–8 days old	5–8 days	This is the standard age used in WHO tunnel tests [6].
Status of mosquito	Unfed	-	-	-	Non-blood-fed; Sugar-starved, 6 h.		Nulliparous. Sugar-starved, 6 h	Nulliparous. Non-blood-fed. Sugar-starved for a minimum of 6 h.	This is the standard mosquito status used in WHO tunnel tests [6]. Consensus agreed sugar-starving found increase mosquito responsiveness to bait.
Mosquitoes per replicate	5	5	50	10	20–25		100	50	Preliminary research has shown no difference between using 50 or 100 mosquitoes in tunnel tests with IG2 (Kamande, Personal communication).
Samples per net	2 (30 × 30 cm)	Baseline: 5 pieces (1 top, 4 sides). Post-baseline: 4 pieces (1 top, 3 sides).	1 piece (position 2), 25 × 25 cm, 9 × 1 cm holes.	-	4 tubes (4 net pieces).		4 (30 × 30 cm)	2 pieces (1 from roof, 1 from sides); 30 × 30 cm, 9 × 1 cm holes in net.	In the standard WHO tunnel test, one net piece is used [6]. The increase allows a 2nd piece from the roof to be tested. During their manufacture, roof panels can come from different net runs than side panels [12].
Replicate tests per piece of net	2	4 replicates	2 replicates	-	1 replicate per net.		?	1 replicate per net piece.	This is the standard used in WHO tunnel tests [6].
Replicate nets per treatment	30	30 nets (timepoint: 0–30 months), 50 nets (timepoint: 36 months).	30 nets (timepoint: 0–30 months), 50 nets (t36).	Sub-set of nets	2 per testing day (200–250 mosquitoes).			A minimum of 30 nets for each treatment at each time point.	WHO guidelines [6] recommend a minimum of 30 nets (at time points 0–24 months), and a minimum of 50 nets at 36 months testing.
Storage of netting pieces (prior to testing)	cool dry place at 4°	-	-	-				Refrigerated or in a cool dry place, at <5 °C or as per manufacturer’s instructions.	-
Entomological endpoints measured	KD: 30 min. KD: 60 min. Mortality: 24 h.	KD: 60 min. Mortality: 24 h. Mortality: 48 h. Mortality: 72 h.	KD: 60 min. Mortality: 24 h. Mortality: 48 h. Mortality: 72 h. Blood feeding.	-	KD: 60 min. Mortality: 24 h. Mortality: 48 h. Mortality: 72 h.	Mortality: 24 h. Mortality: 72 h. Net penetration. Blood feeding. Blood feeding inhibition. Corrected mortality due to chlorfenapyr.	Collection compartment. Blood-feeding status. ‘Immediate’ mortality (07:00). ‘Delayed’ mortality 24 h, 48 h, 72 h.	Collection compartment. Blood-feeding status. Mortality on collection (‘immediate’). 24 h, 48 h, 72 h mortality (‘delayed’).	These endpoints are sufficient to capture the efficacy of a pyrethroid + CFP net.
Other		Cone test is only looking at impact of alphacypermethrin.	18:00: introduced; 08:00: end.	-	Conducted in darkness during the ‘night phase’ of mosquitoes’ circadian rhythm; 27 ± 2 °C and 75% ± 10% relative humidity. Acclimatized to holding tubes for 1 h.		18:00: introduced; 07:00: end. Conducted in darkness, 27 ± 2 °C and 75% ± 10% relative humidity. Mortality corrected for alpha mortality.	Conducted in darkness during the ‘night phase’ of the mosquitoes’ circadian rhythm. Blood meal source preferably the same as what was used to feed the strain in colony, 27 ± 2 °C and 75% ± 10% relative humidity.	Higher mortalities have been observed when chlorfenapyr is used overnight [16], when, as a result of the *Anopheles* circadian rhythm, flight is increased, and, subsequently, cellular respiration and oxidative metabolism, which the chlorfenapyr targets ([17]), is at its peak.

## Data Availability

Data is contained within the article or Appendix A.

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
