# Peer review of "Developing Consensus Standard Operating Procedures (SOPs) to Evaluate New Types of Insecticide-Treated Nets"

_insects, 2021, doi:10.3390/insects13010007_

Round 1

Reviewer 1 Report

The different MoAs of the new insecticides necessitate the need for new protocols to reliably measure the biological durability of our so-called ‘next-generation’ nets. The authors present a set of standardized methods to measure how long these nets remain active against mosquitoes, which allows for the evaluation of their long-term efficacy. This is a well-written paper with clear and useful SOPs. I only have a few minor questions for the authors.

1) PYR+PBO:

- How useful is it to compare PYR+PBO nets and PYR-only nets (two controls), if the PYR bio-availability differs between your controls? Is it worth the extra work/mosquitoes?

- ‘Samples per net’. Two samples from the side of the net in addition to two from the net roof. What is the solution for nets with PBO only in the roof? Or is that solved by sampling roofs from additional nets (which is something that is mentioned in another column)?

- Why not assess delayed mortality here? As we have seen with both IRS and LLIN products, mosquito mortality can increase significantly over time. I imagine that this kind of data will be very helpful for e.g. impact models (and this is not a lot of extra work).

- Line 203, I would argue that the number of adults that are collected needs to be recorded daily (if their offspring is tested). When it comes to larval collections, the number of larvae/pool (and number of pools) has to be recorded every day. This allows us to ensure there was enough genetic variation in the test populations (I now see e.g. 300 larvae being collected from a single pool, and the adults used in susceptibility assays).

2) PYR+PPF:

The timing of blood feeding. Practically, the 3-9h before net exposure means feeding in the morning and doing the exposure during the afternoon. Why this wide range of hours? Biologically, do you think a female mosquito takes a bloodmeal elsewhere and contacts the PYR+PPF net 3-9h later? It seems to me that a female mosquito contacts the net while successfully feeding on the person sleeping under it (so exposure and feeding happen simultaneously). I am just wondering about gene expression here (we know many genes are up/down-regulated during feeding, but now there is an exposure happening simultaneously).   

3) I appreciate the fact that circadian rhythm is taken into account in the PYR+CFP tests, but why not add a similar recommendation when testing the other two net types? One can either conduct the tests at night, or modify the light-dark cycle in their insectaries. I expect more mosquito activity during the ‘night phase’, which may affect the efficacy of the other net types as well.

Author Response

1) PYR+PBO:

- How useful is it to compare PYR+PBO nets and PYR-only nets (two controls), if the PYR bio-availability differs between your controls? Is it worth the extra work/mosquitoes?

  • As stated in the SOP it is preferential to use a PYR-net with the same characteristics as the PYR+PBO net. However, we appreciate that the feasibility of testing this is challenging due to differing net specification of manufactured nets. We agree with the reviewer that this is an ‘imperfect’ comparison, however it will still provide interesting data on the differences of net actually used in the field. A this is a cone test (using n = 5 mosquitoes per rep), the numbers required are relatively small and it equates to little added work.

- ‘Samples per net’. Two samples from the side of the net in addition to two from the net roof. What is the solution for nets with PBO only in the roof? Or is that solved by sampling roofs from additional nets (which is something that is mentioned in another column)?

  • Samples remain the same, however it is suggested (and clarified in the SOP) that resistant strains are only tested on the PYR+PBO panels:
    • For a pyrethroid-only test net panel: Use pyrethroid-susceptible mosquito strains
    • For a pyrethroid + PBO test net panel: Use pyrethroid-susceptible and pyrethroid-resistant mosquito strains

- Why not assess delayed mortality here? As we have seen with both IRS and LLIN products, mosquito mortality can increase significantly over time. I imagine that this kind of data will be very helpful for e.g. impact models (and this is not a lot of extra work).

  • Questions regarding delayed mortality while of interest, would be more beneficial when evaluating net efficacy at baseline. Collecting delayed mortality data (for this number of nets i.e. 30+, at multiple time point i.e. 6,12,18,24,26 months), would put a significant strain on testing facility, specifically for durability studies where multiple net types may be being evaluated at once

- Line 203, I would argue that the number of adults that are collected needs to be recorded daily (if their offspring is tested). When it comes to larval collections, the number of larvae/pool (and number of pools) has to be recorded every day. This allows us to ensure there was enough genetic variation in the test populations (I now see e.g. 300 larvae being collected from a single pool, and the adults used in susceptibility assays).

  • The protocol does not suggest testing F1 offspring of collected females. Larval collected mosquitoes should only be conducted when a suitable well characterised strain is unavailable. It is already specified that larval pool data should be recorded: “For mosquitoes collected as larvae from the field, details on collection procedure such as the number and distribution of collection sites, and mosquito rearing conditions, should be recorded.”

2) PYR+PPF:

The timing of blood feeding. Practically, the 3-9h before net exposure means feeding in the morning and doing the exposure during the afternoon. Why this wide range of hours? Biologically, do you think a female mosquito takes a bloodmeal elsewhere and contacts the PYR+PPF net 3-9h later? It seems to me that a female mosquito contacts the net while successfully feeding on the person sleeping under it (so exposure and feeding happen simultaneously). I am just wondering about gene expression here (we know many genes are up/down-regulated during feeding, but now there is an exposure happening simultaneously).   

  • The 3-9 hours reflects what the group consensus agreed was a feasible timeframe which was a compromise between a short time range (to decrease potential variability) and what was considered logistically possible within as many studies sites as possible. The aim of these SOPs is to enable us to detect if there is a change in the entomological endpoints overtime (which we can relate to durability of the product), therefore the experimental parameters chosen are those which we think will provide us the best changes of being able to detect this difference.

3) I appreciate the fact that circadian rhythm is taken into account in the PYR+CFP tests, but why not add a similar recommendation when testing the other two net types? One can either conduct the tests at night, or modify the light-dark cycle in their insectaries. I expect more mosquito activity during the ‘night phase’, which may affect the efficacy of the other net types as well.

  • Adding these recommendation may make scheduling of testing more difficulty (particularly in sites where multiple testing for multiple project is conducted). There is evidence which suggest this has a sig. impact in the PYR+CFP nets which is why it was important to include it here and not for the other net types.

Reviewer 2 Report

This is a useful SOPs for evaluating and comparing the three types of pyrethroid mixture-treated nets. The manuscript should be considered to be published after a minor revision. The following comments and suggestion may assist the authors for further revision. 

  1. Spelling out the SOP in the title.
  2. Line 158. Give the six pyrethroids.
  3. Table 2. Give mosquito species names in the table 2.
  4. Line 239-240. "Change 3 replicates to 2 replicates due to short of mosquitoes". My person suggestion is to keep the 3 replicates. You may consider to reduce the number of mosquitoes per replication.
  5. Have you consider any impact and requirement for net materials?
  6. In the discussion, probably you need to mention about training for the SOP and make sure that every participates are at the same page and do the identical work.
  7. Have you consider about other active ingredients for the new insecticide-treated bed netting development? There are some reports about the mixture with attractive toxic sugar baits, repellents, and BTi... 

Author Response

  1. Updated to standard operating procedure
  2. Updated to list the 6 pyrethroid + PBO net brands
  3. Updated table to include species names
  4. Line 239-40 relates to the 4 samples being taken from the net (2 from roof, 2 from sides). This provides a good number of technical replicates per nets and was agreed by group consensus to be a feasible number for testing.
  5. These SOPs are to provide guidance for studies evaluating the durability of next-gen nets. Nets would already be available as part of these tests. Number of control nets would be minimal.
  6. Texted added (lines 441 – 444): “At this stage, it is imperative to ascertain that the methods can be implemented and used successfully within research teams, an identify training needs, if required. This is to ensure that data collected using these methods are as transferable and comparable as possible.”
  7. This publication specifically focused on the development of SOPs for new net types currently being deployed, due to the urgency of needing methods for measuring their durability. Under the auspices of the Innovation to Impact programme we will be considering other vector control methods which may require consensus SOPs or external validation.